# The Psychosocial Impact of Polycystic Ovary Syndrome

**Virginie Simon [1], Maëliss Peigné [2,3] and Didier Dewailly [4],\***

[1] Department of Assisted Reproductive Technologies and Fertility Preservation, Jeanne de Flandre Hospital, 59000 Lille, France

[2] Department of Assisted Reproductive Technologies and Fertility Preservation, Jean Verdier Hospital, 93140 Bondy, France

[3] Faculty of Medicine, University of Sorbonne Paris Nord, 93000 Bobigny, France

[4] Faculty of Medicine, University of Lille, 59000 Lille, France

\* Correspondence: didier.dewailly@orange.fr

**Abstract:** Polycystic ovary syndrome is a common endocrine disorder affecting 5 – 20% of women in association with metabolic disorders and insulin resistance. Patients with PCOS are also at increased risk of developing cardiovascular sound aspects of polycystic ovaries and metabolic complications, a psychosocial impact that exists, which is poorly known, assessed and treated. The delay, sometimes long, for diagnosis and its announcement has a strong impact on the feelings and life projects of these patients. Psychological co-morbidities such as depression, anxiety, eating disorders as well as a decrease in self-esteem and quality of life are frequently described in these patients and must, therefore, be screened and treated.

**Keywords** PCOS; psychological impact; depression; anxiety; quality of life



## 1. Introduction

Polycystic ovary syndrome (PCOS) is a common endocrine disorder affecting 5–20% of the female population [1]. PCOS is the leading cause of female infertility due to anovulation. It is defined by the Rotterdam criteria [2]. Its diagnosis is made when at least two of the following three criteria are met: cycle disorders, hyperandrogenism (clinical and/or biological) and the aspect of polycystic ovaries on ultrasound, after exclusion of other causes of cycle disorders or hyperandrogenism. PCOS is often associated with metabolic disorders and insulin resistance. Patients with PCOS are also at increased risk of developing cardiovascular disease, carbohydrate intolerance and type 2 diabetes [3]. The pathophysiology of polycystic ovary syndrome has long been considered as an ovarian pathology and is still poorly understood. Its medical management, in the context of a desire for pregnancy or not, is well codified and governed by international recommendations [2,4]. Although it is quite common in the general population, a large proportion of patients with PCOS are diagnosed late during a fertility check-up or suffer from diagnostic erraticism, delaying the age of diagnosis and resulting in a lack of information. The psychosocial impact, which is often not well known by the practitioners who have had to deal with it, requires just as much attention as cycle disorders, hyperandrogenism or infertility, which are often the focus of its management. Indeed, although it is less classically reported in the complications of PCOS, patients with PCOS are at greater risk of depression and psychiatric disorders [5]. In the international recommendations published in 2018 and 2020, psychological comorbidities such as depression, anxiety and decreased quality of life should be evaluated and considered in the management of PCOS, and even more when obesity or overweight is associated [4,6].

## 2. A key step: PCOS Diagnosis

Data exist on the impact of the diagnosis of PCOS in patients with PCOS. If the delay in the diagnosis can initially induce a form of relief at the time of diagnosis, its establishment can also be a source of anxiety and stress for the patients, particularly concerning their life plans and the announcement of possible infertility at the time of diagnosis [7]. About the diagnosis of PCOS, several factors have been identified as having a significant psychological impact on patients: the delay and barriers to diagnosis, a lack of empathy from the medical profession, difficulties in accessing a PCOS specialist and the lack of information at the time of diagnosis on the pathology [8]. The diagnosis of PCOS can also be difficult for the practitioners themselves according to a qualitative study based on semi-structured interviews conducted with 36 Australian practitioners and published in 2020 [9]. Indeed, difficulties in diagnosis seem to be based on the risk of over-diagnosing PCOS, which should remain a diagnosis of exclusion. Practitioners also questioned the effect of the announcement of such a diagnosis: positive on certain aspects (understanding of the various symptoms, possible modifications of lifestyle) but also recognize negative effects (stigmatization of patients, stress induced by the diagnosis).

### 2.1. Time to Diagnosis

A very recent study published in 2021 compared, in a cohort of Australian patients, three groups of patients in a guided interview format [10]. The first group had a previous diagnosis of PCOS, the second group included patients who were diagnosed with PCOS during the study and the third group included patients without PCOS. Among the 56 patients who were diagnosed before the study, the mean age of diagnosis was 24.4 years. These patients had more cycle disorders, acne and hirsutism. These patients had more often experienced difficulties in achieving pregnancy. Nevertheless, live birth rates were not significantly different between the three groups. The mean age of patients in the group diagnosed during the study was 30.4 years. Of the patients with PCOS, 50%, whether previously diagnosed or diagnosed during the study, had signs of depression. Time to diagnosis is independently associated with depression and anxiety in patients with PCOS [11] and even if the criteria are well codified, confirmation of the diagnosis can sometimes be lengthy. Indeed, in a study published in 2017, nearly half of the participants had met at least three health professionals before the diagnosis of PCOS was made [12]. In this study including 1385 patients with PCOS, time to diagnosis and the number of practitioners seen before diagnosis were negatively correlated with patient satisfaction.

### 2.2. Information Given at Diagnosis

Information given at diagnosis of PCOS is generally not satisfying for patients. In two cross-sectional questionnaire studies including 210 and 1385 patients [12,13], a lack of information especially about long-term complications of PCOS and the need for psychological support were reported. An Australian study published in 2020 examined the experiences of 10 obese or overweight patients with PCOS at the time of diagnosis [14].

Five main negative points to improve were reported in this study: the feeling of a "PCOS package" (all internal and external manifestations are linked and not necessarily all sought out or considered by the staff caring for them), the lack or absence of information at diagnosis and latency of time to diagnosis, isolation and need for support groups, lack of curative treatment or interest in it, and long-term consequences of PCOS as depression, cardiovascular pathologies and diabetes.

A French study analyzed hundreds of comments posted by infertile patients with PCOS on Internet forums. Patients regularly used these forums, or the Internet in general, to compensate for the lack of information received in consultation. They felt isolated facing the diagnosis, although the information found on the Internet can be a source of anxiety for some of them [15].

### 2.3. Impact of PCOS Diagnosis

The potential delay in diagnosis in patients with PCOS appears to have potentially negative consequences for these patients. However, the reactions to the announcement and the impact of a PCOS diagnosis also raise some questions. Indeed, some studies have shown that patients with PCOS had significantly more anxiety disorders and significantly lower self-esteem than non-diagnosed patients [5,16].

Thus, the announcement of the diagnosis of PCOS seems to have positive as well as negative effects [7]. A randomized study including 181 patients and published in 2017 [17] evaluated the psychological impact of a PCOS diagnosis and the impact of this one on the willingness to undergo or not an ultrasound in Australian university students. Participants were asked to imagine that they had experienced cycle disorders in the previous 6 months as well as acne and hirsutism. Participants were then randomized into four groups according to whether they had been diagnosed with PCOS (yes or no) and whether they were informed that ultrasound was unreliable in diagnosing PCOS in very young patients (yes or no). The median age of the participants was 19 years. Self-esteem was significantly lower and the sense of severity of the condition significantly higher in the PCOS diagnosis group. When patients were informed of the risk of overdiagnosis, particularly with pelvic ultrasound, willingness to undergo ultrasound and feelings of severity were significantly lower in this subgroup. The diagnosis of PCOS also has an impact on the well-being of these patients, their behavior and the lifestyle choices they make. Copp et al. highlighted different feelings upon diagnosis: relief, shock, worry, resilience and even indifference [7]. The fear of being infertile and the fear of developing some pathology such as diabetes were frequently found. The diagnosis of PCOS also had consequences on the use of contraception, on the planning of a possible pregnancy and a possible "pressure" on the couple, as well as on the decision whether or not to terminate an undesired pregnancy if this was their only chance to conceive. The diagnosis also leads to lifestyle changes (diet, physical activity, etc.), although these are sometimes difficult to maintain over time. Health professionals must therefore be attentive to the psychological consequences of the diagnosis of PCOS [18].

## 3. Quality of Life in Patients with PCOS

Patients with PCOS have significantly lower quality of life scores than patients without PCOS [19,20]. Obesity, hyperandrogenism, PCOS-related complications and depression seem to have a negative impact on the quality of life of these patients, although they are not systematically managed [21]. However, their management seems to improve the quality of life of these patients even if it remains inferior to the control groups [19]. In a study evaluating the effect of oral contraception in patients with PCOS suffering from oligospaniomenorrhea and hirsutism, the authors found an improvement in the patients' well-being in relation to the improvement in hirsutism and cycle disorders [22].

However, the use of contraception did not have a positive impact on depression and anxiety scores after 6 months of treatment. The negative impact of PCOS on quality of life seems to last over time and even in the last years of reproductive life as studied in a Northern Finland cohort including the same patients at the age of 31 and 46 years, in the same way as in some chronic pathologies such as asthma or migraine [23]. It is now recommended to assess the impact of PCOS on quality of life with the help of questionnaires such as the PCOS, which was developed specifically for the assessment of quality of life in patients with PCOS and assesses 5 domains (weight, cycle disorders, hirsutism, infertility and psychological impact) with 26 questions [24].

### 3.1. Impact on Body Image

Patients with PCOS appear to have more concerns about their weight and less appreciation and evaluation of their appearance and health [11]. These factors are predictive of a higher risk of depression and higher anxiety scores. Williams et al. [25] conducted video

interviews with 10 patients with PCOS who had a variety of manifestations (cycle disorders, hirsutism, acne, obesity, etc.). Six of them reported an impact of PCOS on their identity as a woman and expressed the extent to which PCOS-related symptoms can threaten their self-perception as a woman. In a study using a photographic analysis method reflecting the experience of patients with PCOS [26], masculine vocabulary or references were used by patients to describe their symptoms, for example, "shaving the moustache".

### 3.2. Impact on Lifestyle

Lifestyle modifications, especially when there are associated with being overweight or obesity, are an integral part of the management of PCOS and represent the first line of non-pharmacological management of PCOS [4,6]. In a study evaluating diet, including vegetable intake and increased physical activity in newly diagnosed patients with PCOS, the authors did not observe any change after the diagnosis of PCOS [27]. These lifestyle modifications (dietary measures, physical activity), although they have been shown to improve certain parameters of PCOS (ovulation rate, cycle regularity), are difficult to show a positive effect on the quality of life of these patients because only a few data exist in this area [28]. Patients with PCOS also appear to be at greater risk of weight gain than patients without PCOS. This weight gain seems to be associated with lifestyle habits (diet, physical activity) [29].

### 3.3. Impact on Sexual Life and the Use of Contraception

The diagnosis of PCOS impacts the use of contraception. In a study published in 2020, 14% of patients stopped their contraception after the diagnosis of PCOS (compared to 4% in patients without PCOS) [27]. An Australian cohort published in 2014 found a lower proportion of patients using contraception in patients with PCOS, although the oral contraceptive pill is one of the keys to the management of cycle disorders and hyperandrogenism in these patients [5]. These patients were also more likely to be trying to conceive. Furthermore, in patients without a desire for pregnancy, the authors found lower rates of contraceptive coverage [30]. PCOS also appears to have an impact on patients' sexual life and satisfaction, regardless of the presence of infertility [31]. In a study comparing patients with and without PCOS using interviews and validated self-questionnaires, the authors found a significantly higher proportion of patients who were dissatisfied with their sexual life and a higher proportion of insufficient vaginal lubrication in patients with PCOS [32]. In contrast, the number of partners and frequency of sexual intercourse did not seem to differ [32,33]. Hirsutism, which is one of the diagnostic criteria for PCOS, seems to have an impact on sexuality in patients with PCOS [33].

PCOS also has an impact on sexual life in adolescents [34,35]. They reported significantly less sexual intercourse than control patients, although the age of first intercourse was not significantly different between the two groups [35]. A review published in 2022 found an impact of PCOS on the sexual health of patients in several areas (desire, lubrication, satisfaction, pain, etc.) [36]. The sexual health of these patients should, therefore, be assessed in consultation.

### 3.4. Impact on Life Plans/Role of Infertility

Patients with PCOS have concerns about their ability to conceive from adolescence [34,35]. A study published in 2018 [37] recruited 30 patients with PCOS through an advertisement on the social network Facebook. Several themes emerged from this study: concern about being able to conceive, lack of information regarding guidance for these patients considering pregnancy and a desire to diversify sources of information regarding PCOS and infertility. A majority of patients with PCOS are aware of possible PCOS-related infertility and express anxiety about that. Nevertheless, in a study published in 2008 [31], the authors did not find a significant difference in depressive symptoms or quality

of life evaluation between patients with PCOS with a current desire for pregnancy and those without a desire for pregnancy.

## 4. Psychological Impact of PCOS

The Androgen Excess—Polycystic Ovary Syndrome Society published recommendations in 2018 by conducting a literature review and meta-analysis on four conditions: anxiety, depression, quality of life and eating disorders [20]. Given the prevalence of these psychological disorders in the population with PCOS, it is recommended to systematically investigate these disorders during management.

### 4.1. Depression and Anxiety

Patients with PCOS have significantly more depressive disorders and anxiety disorders [16,20,38–40]. Indeed, these patients have three times more depressive symptoms and five times more anxiety disorders than patients without PCOS [39]. Although patients with PCOS suffering from depression have a higher body mass index (BMI) and more hirsutism, the significant increase in depressive disorders in patients with PCOS persisted after adjustment for BMI [11,39].

In some studies, such as Fernandez et al. [10], the prevalence of signs of depression reaches almost 50% of patients with PCOS (whether diagnosed at the time of the study or known prior to it). However, it is difficult to assess whether the depressive and anxiety disorders found in patients with PCOS are a consequence of the diagnosis of PCOS and the sometimes-stigmatizing factors associated with it (obesity, hirsutism, infertility) or whether they are inherent to the PCOS [38]. The presence of PCOS in the mother could also be at the origin of the development of psychiatric disorders in children born to mothers with PCOS. Indeed, in a recent study published in 2020 [41], maternal PCOS was associated with the development of psychiatric disorders such as mood disorders, anxiety disorders, sleep disorders, autistic disorders and developmental disorders, and even more when the BMI increased, suggesting a probable synergistic effect of obesity and PCOS.

### 4.2. Bipolar Disorder and Schizophrenia

The evidence for an increased risk of bipolar disorder or schizophrenia in patients with PCOS is inconsistent [40,42] and, therefore, no link between PCOS and these psychiatric disorders can be concluded.

### 4.3. Eating Disorders

Obesity and body image dissatisfaction are known risk factors for eating disorders (ED) [43]. Although this notion remains controversial, the prevalence of eating disorders seems to be higher in the population with PCOS [20,44]. An Australian longitudinal study assessing self-esteem, prevalence of ED and psychological distress in patients with PCOS, using validated scales and patient-reported data, found a significant increase in the prevalence of diagnosed or treated ED in these patients, even after adjusting for some confounding variables such as BMI [45].

### 4.4. Psychological Impact of PCOS Treatment

Little data exists on the impact of PCOS treatments on the psychosocial impact of PCOS. Hirsutism is one the characteristic features of PCOS and may lead to depression and anxiety in these patients [19]. A randomized trial evaluated the impact of laser treatment on psychological morbidity in women with PCOS [46]. In this study, laser treatment improved depression and anxiety scores in the intervention group but there was no significant difference for self-esteem scores.

Lifestyle management could also improve quality of life of obese women with PCOS but only a few data exist [28]. A randomized trial published in 2009 showed that restricted diet could improve depression and quality of life score for patients with PCOS. In this

study, exercise did not provide any additional benefit [47]. The authors also found that weight loss improved depression scores only in the short-term but the positive effect was not observed in the long-term. More recently, in a review published in 2021 [48], the authors showed that exercise is effective for improving health-related quality of life in patients with PCOS and could also reduce anxiety and depression symptoms.

Metformin could also have a role [49,50]. Indeed, use of metformin is associated with a significant decrease of testosterone levels, insulin resistance and menstrual disorders. It also has a positive impact on psychological aspects of quality of life [49]. Despite the improvement of quality of life after treatment, the scores remained decreased compared to control patients. In this study, use of metformin also improved sexual satisfaction.

## 5. Conclusions

Although the psychosocial impact of PCOS is significant, practitioners caring for patients with PCOS know little. Screening for psychological co-morbidities, related or consecutive to PCOS, is essential in order not to ignore psychological suffering or any impact of PCOS on quality of life. It is worth implementing in these patients, already at the PCOS diagnosis stage, a questionnaire describing the psychosocial status of selected PCOS patients. It will certainly speed up the diagnosis and self-assessment of the patient, also directing her to a psychologist or psychiatrist to start psychological and psychiatric therapy. The global management of the PCOS patient is essential throughout her life by the different health actors, both on the physical and psychological levels, one and the other being intimately linked. Indeed, it is increasingly demonstrated, especially in patients with PCOS, that quality of life plays a role in the management of one's health, especially one's lifestyle, and vice versa. Information about the disease needs to be improved among the general public and the medical community.

**Funding:** This research received no external funding.

**Institutional Review Board Statement:** Not applicable.

**Informed Consent Statement:** Not applicable.

**Data Availability Statement:** Not applicable.

**Conflicts of Interest:** The authors declare no conflict of interest.

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
