# Peer review of "The Psychosocial Impact of Polycystic Ovary Syndrome"

_2673-3897, doi:10.3390/reprodmed4010007_

Round 1

Reviewer 1 Report

Polycystic ovary syndrome is a common endocrine disorder affecting 5-20% of women in reproductive age and is the leading cause of anovulation and female infertility. Beside usual symptoms, such as cycle disorders, hyperandrogenism, ultrasound aspects of polycystic ovaries and metabolic complications, a psychosocial impact exists which is poorly known, assessed and treated. The authors reviewed the psychological co-morbidities such as depression, anxiety, eating disorders as well as a decrease in self-esteem and quality of life. Overall, this article focused on an interesting topic of psychosocial impact of PCOS. Here are the specific comments on this article:

1.      It would be better if the authors draw an illustration of the psychosocial impact of PCOS.

2.      The psychosocial impact of the treatment of PCOS was not mentioned in this article.

Author Response

Dear Editor, Dear reviewers,

We would like to thank you for reviewing our manuscript entitled « THE PSYCHOSOCIAL IMPACT OF POLYCYSTIC OVARY SYNDROME: a literature review » for publication in the category «commentary» in the journal Reproductive Medicine.

Please find below our responses and modifications to the manuscript.

Reviewer n°1 :

Polycystic ovary syndrome is a common endocrine disorder affecting 5-20% of women in reproductive age and is the leading cause of anovulation and female infertility. Beside usual symptoms, such as cycle disorders, hyperandrogenism, ultrasound aspects of polycystic ovaries and metabolic complications, a psychosocial impact exists which is poorly known, assessed and treated. The authors reviewed the psychological co-morbidities such as depression, anxiety, eating disorders as well as a decrease in self-esteem and quality of life. Overall, this article focused on an interesting topic of psychosocial impact of PCOS. Here are the specific comments on this article:

  1. It would be better if the authors draw an illustration of the psychosocial impact of PCOS.

Our paper being a commentary, we are not sure as to whether we can include a figure. Furthermore, we do not see precisely what you mean by « illustration of the psychosocial impact of PCOS”.

  1. The psychosocial impact of the treatment of PCOS was not mentioned in this article.

Thank you for your comment. It’s true we didn’t mention the impact of PCOS treatment. Little data exist in the literature but we added a paragraph about this in the text.

Reviewer 2 Report

Polycystic ovary syndrome is a common endocrine disorder. It is also associated with infertility, which can lead to psychological and psychological disorders in the form of depression, apathy and fears. Therefore, it is very important to thoroughly investigate psychosocial symptoms, which are still not fully understood in polycystic ovary syndrome. This work will contribute to a better understanding of the psychosocial problems of PCOS. It is worth implementing in these patients, already at the PCOS diagnosis stage, a questionnaire describing the psychosocial status of selected PCOS patients. It will certainly speed up the diagnosis and self-assessment of the patient, also directing him to a psychologist or psychiatrist to start psychological and psychiatric therapy. Very interesting manuscript

Author Response

Reviewer n°2 :

« Polycystic ovary syndrome is a common endocrine disorder. It is also associated with infertility, which can lead to psychological and psychological disorders in the form of depression, apathy and fears. Therefore, it is very important to thoroughly investigate psychosocial symptoms, which are still not fully understood in polycystic ovary syndrome. This work will contribute to a better understanding of the psychosocial problems of PCOS. It is worth implementing in these patients, already at the PCOS diagnosis stage, a questionnaire describing the psychosocial status of selected PCOS patients. It will certainly speed up the diagnosis and self-assessment of the patient, also directing him to a psychologist or psychiatrist to start psychological and psychiatric therapy. Very interesting manuscript »

We thank for your comment. We added the sentence you proposed in the conclusion.

Please accept the assurance of our sincere consideration.

The authors.